# **Comparative assessment of GRASP algorithm for a dust event over Granada (Spain) during ChArMEx-ADRIMED 2013 campaign.**

J. A. Benavent-Oltra<sup>1,2</sup>, R. Román<sup>1,2</sup>, M. J. Granados-Muñoz<sup>1,2</sup>, D. Pérez-Ramírez<sup>1,2</sup>, P. Ortiz-Amezcua<sup>1,2</sup>, C. Denjean<sup>3</sup>, A. Lopatin<sup>4</sup>, H. Lyamani<sup>1,2</sup>, B. Torres<sup>4,5</sup>, J. L. Guerrero-Rascado<sup>1,2</sup>, D. Fuertes<sup>5</sup>, O. Dubovik<sup>4</sup>, A. Chaikovsky<sup>6</sup>, F. J. Olmo<sup>1,2</sup>, M. Mallet<sup>3</sup>, and L. Alados-Arboledas<sup>1,2</sup>

<sup>1</sup>Department of Applied Physics, University of Granada. 18071, Granada, Spain. <sup>2</sup>Andalusian Institute for Earth System Research (IISTA-CEAMA), University of Granada, Autonomous Government of Andalusia. 18006, Granada, Spain.

Correspondence to: Jose Antonio Benavent-Oltra (jbenavent@ugr.es)

Abstract. In this study, vertical profiles and column-integrated aerosol properties retrieved by GRASP (Generalized
 Retrieval of Atmosphere and Surface Properties) algorithm are evaluated with in-situ airborne measurements made during the ChArMEx-ADRIMED field campaign in summer 2013. In the framework of this campaign, two different flights took place over Granada (Spain) during a desert dust episode on 16<sup>th</sup> and 17<sup>th</sup> June. GRASP algorithm combining lidar and sunsky photometer data measured at Granada, has been used for retrieving aerosol properties. Two sun-photometers datasets are used: one co-located with the lidar system and a second one in the Cerro Poyos station, approximately 1200 m higher
 than the lidar system, but at a short horizontal distance.

- Column-integrated aerosol microphysical properties retrieved by GRASP are compared with AERONET products showing a good agreement. Differences between GRASP retrievals and airborne extinction profiles are in the range 15% to 30%, depending on the instrument onboard the aircraft used as reference. On 16<sup>th</sup> June, a case where the dust layer was coupled to the aerosol layer close to surface, the total volume concentration differences between in-situ data and GRASP retrieval
- are 15% and 36% for Granada and Cerro Poyos retrievals, respectively. On the other hand, on 17<sup>th</sup> June the dust layer was decoupled from the aerosol layer close to surface, the differences are around 17% for both retrievals. In general, all the discrepancies found are within the uncertainly limits, showing the robustness and reliability of GRASP algorithm. However, the better agreement found for Cerro Poyos retrieval with the aircraft data and the vertical homogeneity of certain properties retrieved with GRASP, such as the scattering Angström Exponent, for cases with aerosol layers characterized by different aerosol types show that uncertainties in the vertical distribution of the aerosol properties have to be considered.
  - **1** Introduction

35

5

and terrestrial radiation and to their role in clouds development and precipitation (Boucher et al., 2013). Uncertainties associated with the interaction of atmospheric aerosols with radiation have been reduced in the last years, but there is still a need for improvements mainly in those aspects related to their absorption properties (IPCC, 2013). The characterization of aerosol vertical distribution is another point of interest to reduce uncertainties associated with atmospheric aerosol particles, since they can be different near the surface, within the boundary layer, and in the free troposphere.

Atmospheric aerosols play an important role in the Earth-Atmosphere radiative system due to their interaction with solar

 <sup>&</sup>lt;sup>3</sup>CNRM, Centre National de la Recherche Météorologique (UMR3589, CNRS, Météo-France), Toulouse, France.
 <sup>4</sup>Laboratoire d'Optique Atmosphérique, Université de Lille 1, Villeneuve d'Ascq, France.
 <sup>5</sup>GRASP-SAS, Remote sensing developments, LOA / Université Lille -1, Villeneuve d'Ascq, France.
 <sup>6</sup>Institute of Physics, National Academy of Science, Minsk, Belarus.

Passive remote sensing offers large advances in aerosol characterization with global sun-photometry networks such as the Aerosol Robotic Network (AERONET; Holben et al., 1998) or lunar/star photometry measurements (Pérez-Ramírez et al., 2012; Barreto et al., 2016, 2017). In the last years, different inversion methods, based on spectral aerosol optical depth measurements (AOD), were developed for the retrieval of aerosol microphysical properties such as effective radius (r<sub>eff</sub>)

- and volume concentration (VC) (e.g. Pérez-Ramírez et al., 2015; Torres et al., 2017). Furthermore, other sophisticated algorithms, that use also sky-radiance measurements, were developed for the retrieval of aerosol microphysical properties as well as intensive properties such as single scattering albedo (SSA), asymmetry parameter and aerosol refractive index (RI) (e.g. Nakajima et al., 1996; Dubovik and King, 2000; Olmo et al., 2006, 2008). Nevertheless, all these algorithms/measurements only provide column-integrated aerosol properties.
- Since the 70s, lidar systems are being widely used to characterize aerosol vertical distributions in order to contribute to reduce the radiative forcing uncertainties associated to the atmospheric aerosol. The most basic systems use only information of elastic lidar signal to derive backscatter coefficient by aerosol particles, but require assumption about the extinction-to-backscatter ratio (lidar ratio, LR) (Fernald et al., 1972; Klett, 1981; Fernald, 1984; Klett, 1985). More advance systems such as Raman (Ansmann et al., 1992; Whiteman et al., 1992) and HSRL (High Spectral Resolution Lidar) (Shipley
- et al., 1983; Grund and Eloranta, 1991) are able to provide independent measurements of backscatter and extinction coefficients ( $\beta$  and  $\alpha$ , respectively) without LR assumption. Also, the depolarization measurements are a lidar improvement that provide information about shape of aerosols and allows to characterize the aerosol type (Murayama et al., 2004; Miffre et al., 2011; Bravo-Aranda et al., 2013). However, lidar observations dedicated to the aerosol characterization are very scarce compared to the sun-photometer measurements, and many international networks have emerged in the last decades
- to homogenize and explore such information. This is the case of the global NASA MPLNET network (Micro-Pulse Lidar Network; Lewis et al., 2016) developed for continuous measurements of aerosol and cloud vertical profiles at different sites in the world using standard instrument and data processing algorithms. The EARLINET (European Aerosol Research LIdar NETwork) (Pappalardo et al., 2014) and LALINET networks (Latin American Lidar Network) (Guerrero-Rascado et al., 2016) have also been established in order to provide long term database for the vertical and temporal distribution of
- aerosols over Europe and Latin America, respectively.
  - The retrieval of particle vertical microphysical properties from multiwavelength lidar systems is possible by inverting measurements of three aerosol backscatter and two extinction coefficients, known as the  $3\beta+2\alpha$  configuration, using the algorithms developed by Müller et al. (1999), Böckmann (2001) and Veselovskii et al. (2002). However,  $3\beta+2\alpha$ measurements are scarce compared with the large database of elastic lidar measurements. In this sense, different inversion
- methods were recently developed within the framework of EARLINET in order to retrieve vertical profiles of aerosol microphysical properties using combined information of elastic lidar and sun-photometry measurements. Such kind of approaches were the LIdar-Radiometer Inversion Code (LIRIC; Chaikovsky et al., 2008, 2012, 2016) that provides vertical distribution of volume concentrations, and the Generalized Aerosol Retrieval from Radiometer and LIDAR Combined data (GARRLiC; Lopatin et al., 2013) that also allows the retrieval of SSA and RI. Currently, GARRLiC algorithm is included
- in the Generalized Retrieval of Atmosphere and Surface Properties inversion code (GRASP; Dubovik et al., 2011). However, very few studies have attempted to evaluate this recently developed inversion algorithm (Lopatin et al., 2013; Bovchaliuk et al., 2016; Torres et al., 2017; Román et al., 2017), and therefore their evaluation under different atmospheric conditions is still necessary.
- Field campaigns with state of the art instrumentation offer unique possibilities for the evaluation of new retrievals 40 techniques of particle microphysical and optical properties. Recently, the ADRIMED (Aerosol Direct Radiative Impact on

the regional climate in the MEDiterranean region) field campaign, which was part of the international cooperative research program Chemistry-Aerosol Mediterranean Experiment (ChArMEx; Dulac et al., 2014), was carried out with the main objective of capturing the high complexity of the different aerosol types in the Mediterranean region (Mallet et al., 2016). Several in-situ and remote sensing measurements both from surface and on airborne platforms were collected during this

- 5 campaign using state of the art instrumentation. The measurements were performed at different stations over the western Mediterranean region during summer 2013, to gather an updated database of the physical, chemical and optical aerosol properties as well as the vertical distribution of the major "Mediterranean aerosols" (Mallet et al., 2016; Denjean et al., 2016). Data gathered during ChArMEx-ADRIMED campaign give us an excellent opportunity to evaluate the recently developed algorithms for retrieving aerosol microphysical and optical profiles.
- 10 In that framework, the main objective of this study is to evaluate the aerosol optical and microphysical properties obtained with GRASP during the ChArMEx-ADRIMED field campaign in Granada, Spain. The GRASP configuration evaluated in this study here is the one that combines lidar signals and sun-sky radiance measurements. The paper is structured as follows: Section 2 gives a brief description of the experimental site and the instrumentation employed in this study. GRASP and LIRIC codes are described in detail in Section 3. The results are discussed in Section 4 and, finally, the main conclusions
- are summarized in Section 5.

## 2 Site and Instrumentation

## 2.1 Experimental site

The experimental measurements were obtained over Granada region (Spain) at the Andalusian Institute for Earth System Research (IISTA-CEAMA) of the University of Granada, Spain (37.16° N, 3.61° W, 680 m a.s.l.) and at remote high 20 mountain site Cerro Poyos (37.11° N, 3.49° W, 1820 m a.s.l.), located at Sierra Nevada mountain range about 12 km away (horizontally) from IISTA-CEAMA station. Figure 1 shows a map illustrating the distance between Granada and Cerro Povos stations. The city of Granada is located in south-eastern Iberian Peninsula, and is a non-industrialized medium-sized city with a population around 300,000 (twice including the metropolitan area). The city is sited in a natural basin surrounded by mountains with elevations between 1000 and 3500 m a.s.l.. The area is approximately 200 km from the African continent 25 and approximately 50 km from the western Mediterranean basin. In Granada, one main source of natural aerosol is the long-range transport of mineral dust particles from North Africa (e.g. Lyamani et al., 2005; Valenzuela et al., 2012a) that reaches the area in lofted layers (Müller et al., 2009; Guerrero-Rascado et al., 2008, 2009; Córdoba-Jabonero et al., 2011) before mixing with the atmospheric boundary layer (Bravo-Aranda et al., 2015) and been detected at the surface tor in precipitation samples (Calvo et al., 2010). Another natural source is biomass burning particles: fresh smoke (Alados-30 Arboledas et al., 2011) and long-range transported smoke (Ortiz-Amezcua et. al., 2014, 2017). While the main anthropogenic sources are pollution from Europe, the Iberian Peninsula and the Mediterranean Sea (Lyamani et al., 2006, Pérez-Ramírez et al., 2016), its local sources are mainly road traffic and central heating systems (Lyamani et al., 2012; Titos et al., 2017).

## 2.2 Ground-based instrumentation

35 Columnar aerosol properties during daytime were obtained by CIMEL CE-318-4 (*Cimel Electronique*) sun-sky photometers at IISTA-CEAMA and Cerro Poyos sites. The sun-photometer instruments used in this study are operated in the framework of AERONET - RIMA network (Iberian Network for Aerosol Measurements, federated to AERONET)

(https://aeronet.gsfc.nasa.gov/). A complete description of the instrument can be found in Holben et al. (1998). Briefly, this instrument makes direct solar irradiance measurements at 340, 380, 440, 670, 870 and 1020 nm and sky radiance measurements at 440, 670, 870 and 1020 nm. Solar direct irradiance measurements are used to calculate the AOD at 340, 380, 440, 670, 870 and 1020 nm, with uncertainty of  $\pm 0.01$  for  $\lambda > 400$  nm and of  $\pm 0.02$  for  $\lambda < 400$  nm (Holben et al.,

- 1998; Eck et al., 1999). Furthermore, the Angström Exponent (AE), parameter that describes the spectral dependency of the AOD, is calculated in the range 440–870 nm. AE provides an indication of the particle size: small values (< 0.5) suggest a predominance of coarse particles, while large values (> 1.5) indicate a predominance of small particles (e.g., Dubovik et al., 2002). The solar direct irradiance and sky radiance measurements are used to retrieve aerosol optical and microphysical properties such as columnar particle size distribution (PSD), real and imaginary refractive indices (RRI and IRI) and SSA.
- using the algorithm of Dubovik et al. (2006). In addition, the inversion code provides other variables such as the VC,  $r_{eff}$ and standard deviation for fine and coarse modes of the retrieved PSD. The uncertainty of the AERONET inversion products is described by Dubovik et al. (2000). Briefly, the uncertainty in the retrieval of SSA is ±0.03 for high aerosol load (AOD<sub>440</sub> > 0.4) and solar zenith angle > 50°. For measurements with low aerosol load (AOD<sub>440</sub> < 0.2), the retrieval accuracy of SSA ( $\lambda$ ) drops down to 0.02–0.07 (Dubovik et al., 2000). For high aerosol load and solar zenith angle > 50°.
- errors are about 30%–50% for the imaginary part of the RI. For particles in the size range  $0.1 < r < 7 \mu m$ , errors in PSD retrievals are around 10–35%, while for sizes lower than 1  $\mu m$  and higher than 7  $\mu m$  retrieval errors rise up to 80–100%. In this work, the AERONET Version 2 Level 2.0 data obtained at Granada and Cerro Poyos during ChArMEx-ADRIMED 2013 are used. However, due to the strong limitations imposed by the AERONET inversion algorithm (AOD<sub>440</sub> > 0.4 and solar zenith angle > 50°) there was no SSA and RI AERONET level 2.0 retrievals during the campaign. Thus, for comparing
- AERONET SSA values with GRASP retrievals, the AERONET level 1.5 cloud screened data corresponding to AOD > 0.2and solar zenith angle > 50° are used in this study.

The multi-wavelength Raman lidar MULHACEN, based on a customized version of LR331D400 (Raymetrics S.A.), is used for obtaining vertical profiles of the atmospheric aerosol properties. This system located at Granada, was incorporated to EARLINET in April 2005, being at present a contributing station to ACTRIS research infrastructure (Aerosols, Clouds,

- and Trace gases Research InfraStructure Network) (<u>http://actris2.nilu.no/</u>). The system has a monostatic biaxial configuration alignment, pointing vertically to the zenith and uses a pulsed Nd:YAG laser with 2<sup>nd</sup>- and 3<sup>rd</sup>-harmonic generators, that emits simultaneously pulses at 1064, 532 and 355 nm. The receiving system consists of several detectors, which can split the radiation according to the three elastic channels at 355, 532 (parallel- and perpendicular-polarized; Bravo-Aranda et al., 2013), and at 1064 nm; two nitrogen Raman channels at 387 and 607 nm (shifted signal from radiation
- at 355 and 532 nm, respectively); and a water vapor Raman channel at 408 nm (shifted signal from radiation at 355 nm; Navas-Guzmán et al., 2014). More information can be found in Guerrero-Rascado et al. (2008). The aerosol particle backscatter coefficient profiles obtained from the multi-wavelength lidar were obtained by the Klett-Fernald method (Fernald et al., 1972; Klett, 1981; Fernald, 1984; Klett, 1985). Total uncertainty in the profiles obtained with Klett method is usually 20% for β and 25-30% for α profiles (Franke et al., 2001; Preißler et al., 2011). The procedure suggested by

Wandinger and Ansmann (2002) was applied to the lidar data to correct the incomplete overlap. Without correction, the complete overlap for this instrument is above 1200 m a.g.l. (Navas-Guzmán et al., 2011).

## 2.3 Airborne measurements

During the period from 14<sup>th</sup> June to 4<sup>th</sup> July 2013, 16 flights were performed in the framework of ChArMEx-ADRIMED over the Mediterranean Basin with the ATR-42 aircraft of SAFIRE (French aircraft service for environmental research;

<u>http://www.safire.fr</u>). These flights ascended or descended performing a spiral trajectory during 30 min. Two of these flights (flight number F30 and F31) took place over Granada on 16<sup>th</sup> and 17<sup>th</sup> June 2013, respectively. Figure 1 shows the spiral trajectory of F31 flight that is similar to that of F30, covering in both cases the same atmospheric column. Flight details are described by Mallet et al. (2016) and Denjean et al. (2016).

- 5 Table 1 summarizes the instrumentation onboard the ATR-42 airplane used in this study. The Scanning Mobility Particle Sizer (SMPS) with an accuracy of 5% (Wiedensohler et al., 2012) and the Ultra-High Sensitivity Aerosol Spectrometer (UHSAS) with an accuracy of 10% (Cai et al., 2008) are used for measuring aerosol number size distribution in the submicron range. The wing-mounted Forward Scattering Spectrometer Probe, model 300 (FSSP-300) with an accuracy of 30% (Baumgardner et al., 1992) and the in-cabin GRIMM OPC (sky-OPC 1.129) with an accuracy of 10% (Denjean et al., 2012)
- 10 2016) were used to measure the optical size distributions in the diameter nominal size range between 0.28 and 20 μm and between 0.3 and 32 μm, respectively. The total particle volume concentrations in the diameter range 0.1-30 μm and volume concentrations of fine (0.1-1 μm) and coarse (1-30 μm) modes were calculated from the measured aerosol number size distributions, assuming that aerosol particles are spherical.

In addition, the nephelometer TSI (model 3563) was used to measure particle scattering coefficients at three wavelengths (450, 550 and 700nm) with an accuracy of 5% (Müller et al., 2011) and a Cavity Attenuated Phase Shift (CAPS) was employed to obtain particle extinction coefficient at 530 nm with an accuracy of 3% (Massoli et al., 2010). Also, the PLASMA (Photomètre Léger Aéroporté pour la Surveillance des Masses d'Air) system, which is an airborne sun-tracking photometer, was used to obtain AOD with wide spectral coverage (15 channels between 0.34 – 2.25 μm) with an accuracy of approximately 0.01 (Karol et al., 2013), as well as the particle extinction vertical profiles (Torres et al., 2017).

## 20 3 GRASP and LIRIC inversion algorithms

The input information needed by GRASP and LIRIC algorithms and the aerosol properties retrieved and used in this work are shown in Table 2. LIRIC algorithm provides height-resolved aerosol volume concentration data for the fine and coarse modes from combined lidar and sun-sky photometer information (Chaikovsky et al., 2008, 2012; Granados-Muñoz et al., 2014; Chaikovsky et al., 2016). For this, column-integrated aerosol properties provided by the AERONET code (Dubovik

- et al., 2002, 2006) are used as input, together with the lidar elastic backscatter signals at three different wavelengths (355, 532, and 1064 nm). These data are put through an iterative procedure based on the Levenberg–Marquardt method, which is described in detail in Chaikovsky et al. (2016). Besides the volume concentration, the algorithm retrieves additional data sets, including profiles of particle  $\alpha$  and  $\beta$  coefficients, and LR among others. AERONET column-integrated products used as input are not modified by LIRIC during the retrieval process.
- 30 The GRASP inversion code (Dubovik et al., 2011; Lopatin et al., 2013), was developed at Laboratoire d'Optique Atmospherique (LOA) of the University of Lille. GRASP is based on a similar philosophy than LIRIC code, but goes a step further since it simultaneously inverts both the coincident lidar and sun-sky photometer measurement, retrieving vertical, but also column, aerosol optical and microphysical properties for both fine and coarse modes. The simultaneous inversion of lidar and sun-sky photometer measurements is expected to improve the retrievals since the lidar data
- 35 complement the sky photometer measurement at scattering angles of 180° and the photometer data provides the information (e.g. amount and type) required for lidar retrievals that otherwise would be assumed from climatological data (Bovchaliuk et al., 2016). Therefore, the column aerosol properties obtained by GRASP will differ from the AERONET ones. Additionally, it is worth to note that GRASP allows independently retrieving aerosol optical and microphysical properties for the two distinct aerosol modes, fine and coarse. The retrieval of height-dependent single scattering albedo data is an

additional advantage of GRASP over LIRIC. GRASP also provides an estimation of the systematic and random errors for both the directly retrieved (SD, RRI, IRI, SSA) and derived ( $\alpha$ ,  $\beta$ , VC profiles) aerosol properties. The single scattering albedo profiles errors are not shown because they are unfortunately not provided at the moment. Additional details on GRASP retrieval algorithm and its performance can be found in Lopatin et al. (2013) and Bovchaliuk et al. (2016).

## 5 4 Results

10

15

As we commented before, two of the ATR-42 flights, flight number F30 and F31, performed in the framework of ChArMEx-ADRIMED campaign were carried out over Granada region on 16<sup>th</sup> and 17<sup>th</sup> June 2013, respectively. Figure 2 shows the time series of the lidar range corrected signal (RCS) and the depolarization ratio ( $\delta$ ) at 532 nm on both days measured at Granada station. The RCS is calculated as P\*r<sup>2</sup>, where P is the lidar signal (corrected from background and dark current) and r is the altitude. On the first day, a homogeneous layer is observed from the surface up to 5 km a.s.l., with an elevated aerosol layer coupled to the superficial aerosol layer throughout the day. The next day this layer was decoupled from the aerosol layer close to surface and it disappeared around 13:00 UTC and measurements of  $\delta$  evidence that there was an aerosol type below 2.7 km a.s.l. and another aerosol type above this altitude up to 5.5 km a.s.l.. On 16<sup>th</sup> June, the lidar measurements (marked with purple lines in Fig 2) obtained during the first flight between 14:15 and 14:45 UTC and sun-photometer measurements collected at 16:22 UTC (black dashed line in Fig 2) at Granada and Cerro Poyos were selected for further analysis. The selected sun-photometer measurement was the closest measurement available in time to the first flight. On 17<sup>th</sup> June, the lidar measurements obtained during the second flight (07:15 to 07:45 UTC) and sun-sky photometer measurements obtained at both stations at 07:40 UTC were selected for further analysis.

AERONET products during these flights indicate the presence of dust particles. In fact, on 16<sup>th</sup> June; AOD<sub>440</sub> at 14:15 UTC
was around 0.26 and 0.19 for Granada and Cerro Poyos, respectively, and 0.27 and 0.22 at 16:22 UTC. On this day, the AE<sub>440-870</sub> was 0.30-0.26 (Granada-Cerro Poyos) at 14:30 UTC and 0.34-0.27 (Granada-Cerro Poyos) at 16:22 UTC, indicating moderate atmospheric aerosol load dominated by coarse particles. On 17<sup>th</sup> June, the AOD<sub>440</sub> at 07:40 UTC was 0.21 and 0.18 and the AE<sub>440-870</sub> was 0.43 and 0.30 for Granada and Cerro Poyos, respectively, which also indicates the predominance of coarse particles on this day. The presence of mineral dust over Granada region during both days is confirmed by the analysis of back-trajectories analysis (not shown) by HYSPLIT model (Hybrid Single-Particle Lagrangian Integrated Trajectory; Stein et al., 2015; Rolph, 2016), which indicates that the relevant air masses came from the Saharan region, specifically from Algeria, at different heights.

## 4.1 Comparison of columnar properties retrieved by GRASP and AERONET algorithms

Some of the aerosol columnar properties obtained from AERONET and retrieved by GRASP (combining photometer and
 lidar measurements) on 16<sup>th</sup> and 17<sup>th</sup> June at Granada and Cerro Poyos stations are shown in Figures 3 to 5 and summarized in Table 3.

Figure 3 shows the column-integrated PSD retrieved by both AERONET and GRASP algorithms on 16<sup>th</sup> and 17<sup>th</sup> June for Granada and Cerro Poyos stations. The retrieved PSD evidence the predominance of coarse mode particles, as expected for dust events (Lyamani et al., 2005; Guerrero-Rascado et al, 2009). Both AERONET and GRASP retrieved PSD present

a bimodal behavior, with the radius of fine mode below  $0.5 \,\mu\text{m}$  and the radius of coarse mode above  $0.5 \,\mu\text{m}$ . The differences between the PSD retrieved by GRASP and AERONET are mostly within uncertainties associated with both methods (±10 – 35% for the size range from 0.1  $\mu\text{m}$  to 7  $\mu\text{m}$  and ±35 – 100% outside this range; Dubovik et al., 2000) except for the size

range 5 - 8.7  $\mu$ m where the differences are higher, especially at 6.64  $\mu$ m (> 100%). Furthermore, the coarse mode retrieved by GRASP over both sites shows a clear shift towards higher radii in comparison to the AERONET retrievals (Figure 3). This shift was also observed by Lopatin et al. (2013) during dust and biomass burning events over Minsk, Belarus and by Bovchaliuk et al. (2016) during dust events over Dakar, Senegal. These authors attributed this coarse mode shift towards

higher radii to the use of the lidar data in the GRASP retrievals. The lidar data provide additional information at scattering angles of 180° and further wavelengths compared to the sun-photometer, influencing the size distribution retrieved especially in the coarse mode.

Table 3 summarizes the columnar reff and VC of fine and coarse modes obtained at both stations by AERONET and GRASP algorithms on 16<sup>th</sup> and 17<sup>th</sup> June. The retrieved microphysical properties are similar to those typically obtained during

- African desert dust events over Granada (Valenzuela et al., 2012b). The fine mode reff retrieved by both methods ranges between 0.10 and 0.13  $\mu$ m. Differences between fine mode r<sub>eff</sub> retrieved by GRASP and AERONET are below 0.02  $\mu$ m, which are within the uncertainty of the inversions (Lopatin et al., 2013; Torres et al., 2014). For the coarse mode, the reff values obtained by GRASP were 0.03 µm higher than those retrieved by AERONET but the differences are within the uncertainty range. A similar behavior is observed for the column-integrated volume concentration, with slightly larger 15 values provided by GRASP for the fine and coarse modes (0.016  $\pm 0.003$  and 0.148  $\pm 0.017 \ \mu m^3/\mu m^2$ ) compared to AERONET (0.014  $\pm$ 0.003 and 0.125  $\pm$ 0.013  $\mu$ m<sup>3</sup>/ $\mu$ m<sup>2</sup>), but differences are still within the uncertainties.
- Figure 4 illustrates the retrieved columnar RRI and IRI for each day obtained by GRASP and AERONET at Granada and Cerro Poyos. Moreover, RRI and IRI at 530 nm estimated by Denjean et al. (2016) using airborne measurements over Granada on 16th and 17th June are included in the plot. AERONET provides RRI and IRI for the whole size distribution,
- while GRASP is able to provide RRI and IRI for fine and coarse modes separately. The RRI retrieved by GRASP and AERONET algorithms do not show any spectral wavelength variations, and the differences between RRI values retrieved by both inversion algorithms are within the uncertainties (differences below 5%). Because of the predominance of the coarse mode during the analyzed dust event, both the AERONET and airborne RRI values are close to the values retrieved by GRASP for coarse mode, with differences <0.03, on both days. On the other hand, the IRI values retrieved by GRASP
- for the fine mode present a rather low spectral dependence while IRI values for the coarse mode presents a clear increase in the UV region. These results are coherent with those reported for different absorption species by Schuster et al. (2016) using AERONET data. At Cerro Poyos we did not find the spectral dependence of the IRI typically associated to mineral dust. The AOD at 440 nm were around 0.18 - 0.27 and we used AERONET level 1.5 products, therefore, these values have large uncertainties (> 50%; Dubovik et al., 2000). The lack of spectral dependence can be just an artifact of the inversion.
- However, it is worthy to note that at Cerro Poyos the PSD shows a mode in the coarse mode size range around 1 µm. As there is still discussion in the scientific community about dust RI and about the differences in dust particles between different sources (e.g. Colarco et al., 2014), results can suggest possible differences in dust RI between long range transported and mixture with local dust injections (the area is very dry in summer, thus favoring local mineral dust resuspension) and local pollution. The RRI and IRI values provided by AERONET show good agreement with GRASP
- retrievals for coarse mode, something expected due to the large predominance of dust particles. Better agreement between IRI retrieved by AERONET and by GRASP for coarse mode was found for Cerro Poyos, with differences ~ 10%, while for Granada these differences are between 35 - 80% (larger differences for lower wavelengths). The high discrepancy between IRI retrieved by AERONET and by GRASP in the case of Granada can be explained by the uncertainty associated to the incomplete lidar overlap. Cerro Poyos station is located above the lidar incomplete overlap height, and thus the effect of the incomplete overlap on the retrieval is negligible. This is not the case for the retrieval from Granada station. On 16<sup>th</sup>

June, IRI airborne values estimated at 530 nm are close to IRI retrieved by GRASP at 532 nm for coarse mode, and the differences are within the associated uncertainties. On the other hand, on 17th June, there are more differences between IRI values retrieved by GRASP at Granada station and those estimated from airborne measurement, with differences over 100%, whereas for the Cerro Poyos retrievals the differences are 50%.

- 5 Figure 5 shows the columnar SSA values retrieved by GRASP and AERONET on 16th and 17th June at Granada and Cerro Poyos. Moreover, the SSA value at 530 nm calculated by Denjean et al. (2016) for dust layer using airborne measurements during the campaign was 0.95 ±0.04. SSA retrieved by GRASP at 532 nm are close to the airborne value. Better agreement with this value is found for the retrievals from Granada on 16th June and at Cerro Poyos on 17th June. The differences from Granada on 17th June could be due the in-situ value was calculated for the dust layer whereas that GRASP and AERONET
- use sun-photometer data, which measures the total atmospheric column. Furthermore, in the case of Granada station, these 10 measures could be influenced by injections of local pollution. The retrieved SSA values are in the range between 0.85-0.98 (355-1064 nm wavelength range) and are in the ranges of the typical values for dust aerosols (Dubovik et al., 2002; Toledano et al., 2011; Lopatin et al., 2013). Both AERONET and GRASP retrievals follow the same pattern with wavelength, with increasing SSA as wavelength increases, which is a typical characteristic of dust aerosols (Dubovik et
- 15 al., 2002; Valenzuela et al., 2012b). Differences between SSA retrieved by AERONET and GRASP algorithms are below 0.03 at all wavelengths, within the uncertainties associated with each method. The discrepancies between SSA retrieved by AERONET and GRASP algorithms are obtained for Cerro Poyos station (<1 %) in particular at 1020 nm, whereas for Granada retrievals the differences are bigger and the lowest discrepancies are obtained at 675 nm.

#### 4.2 Comparison of vertical properties retrieved by GRASP and LIRIC algorithms and in situ airborne 20 measurements

Figure 6 shows particle VC profiles for the fine, coarse and total (fine+coarse) modes, retrieved by GRASP and LIRIC, together with the results obtained with airborne instrumentation. Generally, there is good agreement between the profiles retrieved by GRASP and LIRIC and those obtained by the airborne instrumentation, with both retrievals and the airborne data reproducing similar vertical structures on both days. The airborne data show a larger variability compared to GRASP and LIRIC mostly associated to their larger uncertainty and the fact that the airborne data are instantaneous measurements

Table 4 summarizes the volume concentration mean values and associated standard deviations retrieved from the in-situ airborne measurements, GRASP and LIRIC profiles shown in Figure 6. Data are analyzed only for those layers with total volume concentration above 20 µm<sup>3</sup>/cm<sup>3</sup> to avoid undesirable outliers for low aerosol loads. Hence for 16<sup>th</sup> June we analyze

whereas the lidar data are an average over a 30-minute period.

- the layer between 1.2 and 4.5 km a.s.l. and for 17<sup>th</sup> June, the analyzed layer is that from 2.6 to 5.0 km a.s.l.. There is a slight contribution of the fine mode in the dust layers on both days, with values between  $3 \,\mu m^3 / cm^3$  for the airborne data and 5.3  $\mu$ m<sup>3</sup>/cm<sup>3</sup> for LIRIC. In general, it is observed that the coarse mode contributes the most to the total VC, as expected due to the predominance of mineral dust. Coarse mode concentration values range between 28  $\pm$ 4 and 46  $\pm$ 4  $\mu$ m<sup>3</sup>/cm<sup>3</sup> on June16 and 35  $\pm$ 5 and 42  $\pm$ 11  $\mu$ m<sup>3</sup>/cm<sup>3</sup> on June 17, depending on the dataset considered. GRASP and LIRIC retrievals both
- overestimate the airborne data for the fine mode while they overestimate/underestimate the total mode on 16<sup>th</sup>/17<sup>th</sup> June using sun-photometer data at both Granada and Cerro Poyos stations. In the case of fine mode, the differences between the airborne and the retrievals are lower than 5  $\mu$ m<sup>3</sup>/cm<sup>3</sup> (about 80%). The agreement for the coarse mode is high with differences lower than  $6 \,\mu m^3/cm^3$  (25%), except for the LIRIC inversion from Cerro Poyos on 16 June, where the difference is 19  $\mu$ m<sup>3</sup>/cm<sup>3</sup> (around 80%). Both algorithms show the largest differences for the retrievals from Cerro Poyos on 16<sup>th</sup> June, whereas the differences for the retrievals from Granada for total and coarse mode are around 15% and 25% using GRASP

and LIRIC, respectively. On 17<sup>th</sup> June for Granada retrieval, the differences between both algorithms and airborne data below 2 km a.s.l. could be explained because the flight was not exactly over Granada city as shown in Fig. 1 and in the first two kilometers of the atmosphere differences are expected because of the influence of the city. In the dust layer on 17th June, the differences are around 20% for coarse and total volume concentration by both algorithms for Granada and

5 Cerro Poyos stations. Differences between GRASP and LIRIC retrievals are below 30%, well within the combined uncertainty from both retrievals. There are no accurate calculations of the uncertainty associated to LIRIC profiles, but it is estimated to be around 50% in cases of mineral dust (Granados-Muñoz et. al., 2016).

Figure 7 shows the aerosol  $\beta$  coefficient profiles at 355, 532 and 1064 nm retrieved by GRASP and the profiles calculated by Klett-Fernald method. The LR used in Klett method is assumed constant for the entire profile and was computed by

- fitting the integral of the different extinction profiles to the measured aerosol optical depth. However, GRASP uses both sun-sky radiances and the backscatter lidar data to provide LR values, both in column-integrated and vertical profiles. The GRASP LR values are close to the LR values used by Klett-Fernald method, these values are typical from Saharan dust measured over the southeastern Spain (Guerrero-Rascado et al., 2009; Navas-Guzmán et al., 2013). Below 1.6 km, the Klett retrieval at 355 showed unrealistic values probably associated with instrumental problems. However, for GRASP this
- problem does not appear and seems to be canceled due to the use of the combined data of lidar and sun-photometer. GRASP algorithm underestimates the values obtained by Klett-Fernald method, except for Cerro Poyos retrieval on 17<sup>th</sup> June, with larger differences for Granada retrievals. Nevertheless, the differences are within the uncertainties claimed for our system (approximately 30%). The differences at the ultraviolet channel reached 19% and around 9% for Granada and Cerro Poyos retrievals, respectively. The discrepancies between backscatter coefficient profiles at 532 nm retrieved by GRASP and
- Klett-Fernald are around 16% for Granada retrieval on 16<sup>th</sup> June and 11% on 17<sup>th</sup> June and for Cerro Poyos retrievals on both days. In the case of backscatter coefficient profiles at 1064 nm, the differences between both retrievals are close to 24% for Granada on 16<sup>th</sup> June and Cerro Poyos on 17<sup>th</sup> June, while for the other two cases the differences are the lowest (6%).
- The comparison between aerosol α coefficient profiles retrieved by GRASP and those measured by airborne instruments
  (CAPS and PLASMA) is shown in Figure 8. Profiles retrieved by GRASP show good agreement with the CAPS data (measurements only on 16<sup>th</sup> June at 532 nm), even though with slightly higher values for GRASP of approximately 3 ±3 Mm<sup>-1</sup> (7%) and 9 ±5 Mm<sup>-1</sup> (18%) for the inversions from Granada and Cerro Poyos, respectively. GRASP extinction coefficient retrievals were larger than PLASMA measurements at all wavelengths, with larger differences at the ultraviolet channel (~50%). On 16<sup>th</sup> June, the differences for the ultraviolet channel are 20 ±11 Mm<sup>-1</sup> (45%) and for the visible and ultraviolet channels are 11 ±8 Mm<sup>-1</sup> (30% and 40%, respectively). These differences were similar or lower than those obtained by Karol et al. (2013) when comparing PLASMA with lidar data. On 17<sup>th</sup> June, PLASMA and GRASP show the
- same layers, but their differences are larger, reaching 50% for the visible channel and more than 60% for the ultraviolet and infrared channels. As GRASP and LIRIC reproduce the same layer structures for volume concentrations, these differences can be mainly associated with PLASMA.
- 35 Vertical profiles of SSA obtained by GRASP at Granada and Cerro Poyos station on 16<sup>th</sup> and 17<sup>th</sup> June are shown in Figure 9. As SSA is an intensive aerosol parameter, only SSA values for the layers with large aerosol loads are represented. On 16<sup>th</sup> June, there are no remarkable changes in SSA with altitude, which agrees with the extinction and backscatter coefficients profiles and with the particle volume concentrations. For 17<sup>th</sup> June, vertical profiles of SSA are sensitive to the different aerosol layers with different aerosol types illustrating the capabilities of GRASP for detecting different aerosol
- 40 layers with different composition. Nonetheless, the values of SSA are also within those associated with dust aerosol in

previous studies (Dubovik et al., 2002; Toledano et al., 2011; Lopatin et al., 2013). Differences are observed again between the SSA profiles obtained at Granada and Cerro Poyos stations. On 16<sup>th</sup> June SSA differences between both retrievals are lower than 2% while on 17<sup>th</sup> differences reach up to 10%. This result is again associated with overlap issues, although the influence of the city with injection of large amounts of particles confined below the altitude of Cerro Poyos cannot be

neglected in sky radiance measurements.

Figure 10 shows scattering AE computed between 450 and 700 nm,  $AE_{sca}$  (450-700), obtained by GRASP algorithm at Granada and Cerro Poyos stations together with those obtained from nephelometer airborne measurements. GRASP scattering coefficient profiles are calculated by multiplying the extinction coefficient by the SSA at the same wavelength. Despite the fact that the  $AE_{sca}$  (450-700) profiles from the airplane date are noisier than GRASP profiles, general good

- agreement is observed, with discrepancies within the uncertainties. In general, GRASP values are larger than the airborne data for altitudes above 2.5 km a.s.l.. Above this altitude, the  $AE_{sca}$  (450-700) values are close to zero for the airborne data on both days, which is typical of aerosols dominated by coarse particles (Bergstrom et al., 2007). However, in the lower part of the profiles  $AE_{sca}$  (450-700) values are larger (~0.7 and ~1.6 for the airborne data on 16<sup>th</sup> and 17<sup>th</sup> June, respectively) and GRASP profiles underestimate the airborne data. The values for these lower altitudes, including those retrieved by
- GRASP using the sun-photometer data measured at Granada and the airborne data, were similar to in-situ measurements at IISTA-CEAMA, with values around 0.70 ±0.10 and 1.67 ±0.07 on 16<sup>th</sup> and 17<sup>th</sup> June, respectively. GRASP profiles have similar values above and below 2.5 km a.s.l. with better agreement between airborne data and Cerro Poyos retrieval. Granada retrieval shows more differences on 17<sup>th</sup> June, the case with aerosol layers with different aerosol types. On 17<sup>th</sup> June, in the range ~1.8 2.7 km a.s.l. the aerosol load was low (~5 µm<sup>3</sup>/cm<sup>3</sup>) and, hence, SSA and AE values could be less
- reliable in this layer. However, the layer up to 1.8 km a.s.l. showed a concentration moderate (~17 μm<sup>3</sup>/cm<sup>3</sup>) with a different composition to layer above 2.7 km a.s.l. as show the SSA and AE profiles.
  Finally, the Table 5 shows the mean values with ±1 standard deviation of β-AE (532-1064) and color ratio (CR =β(532 nm)/β(1064 nm)) calculated by GRASP in the layer between 1.8 and 4.5 km a.s.l. and between 2.8 and 5.0 km a.s.l. on 16<sup>th</sup> and 17<sup>th</sup> June, respectively. The values of β-AE and CR are 0.5 ±0.2 and 1.3 ±0.3, respectively, are in the

## **5** Summary and conclusion

range of typical values for dust aerosols (Perrone et al., 2014).

GRASP algorithm is applied to lidar and sun-sky photometer measurements at Granada during the ChArMEx-ADRIMED campaign in summer 2013. Data from a second photometer at 1.2 km above the lidar system are also used, locate it above the lidar incomplete overlap height. This second sun-photometer allows us to explore the effect of the lidar incomplete overlap on the retrievals and the influence of the aerosol vertical layering on the results, especially in cases of complex structures when different aerosol types are observed below and above Cerro Poyos. The optical and microphysical properties retrieved by GRASP using independent AERONET data have been compared with airborne measurements corresponding to two flights.

The flights took place on 16<sup>th</sup> and 17<sup>th</sup> June, 2013, during dust events affecting Granada. The GRASP retrievals show a good agreement with AERONET products, with discrepancies well below the uncertainties. Total volume concentration profiles retrieved by GRASP and airborne measurements show a good agreement with differences around 15% on 16<sup>th</sup>June using for the retrieval sun-photometer data measured at Granada and on 17<sup>th</sup>June using for the retrieval sun-photometer data measured at Granada and on 17<sup>th</sup>June using for the retrieval sun-photometer data measured at Granada and on 17<sup>th</sup>June using for the retrieval sun-photometer data measured at Granada and on 17<sup>th</sup>June using for the retrieval sun-photometer data measured at Granada and on 17<sup>th</sup>June using for the retrieval sun-photometer data measured at Granada and on 17<sup>th</sup>June using for the retrieval sun-photometer data measured at Granada and on 17<sup>th</sup>June using for the retrieval sun-photometer data measured at Granada and on 17<sup>th</sup>June using for the retrieval sun-photometer data measured at Granada and on 17<sup>th</sup>June using for the retrieval sun-photometer data measured at Granada and on 17<sup>th</sup>June using for the retrieval sun-photometer data measured at Granada and on 17<sup>th</sup>June using for the retrieval sun-photometer data measured at Granada and on 17<sup>th</sup>June using for the retrieval sun-photometer data measured at Granada and on 17<sup>th</sup>June using for the retrieval sun-photometer data measured at Granada and on 17<sup>th</sup>June using for the retrieval sun-photometer data measured at Granada and on 17<sup>th</sup>June using for the retrieval sun-photometer data measured at Granada and on 17<sup>th</sup>June using for the retrieval sun-photometer data measured at Granada and on 17<sup>th</sup>June using for the retrieval sun-photometer data measured at Granada and 01<sup>th</sup>June using for the retrieval sun-photometer data measured at Granada and 01<sup>th</sup>June using for the retrieval sun-photometer data measured at Granada and 01<sup>th</sup>June using for the retrieval sun-photometer data measured at Gr

lidar data and Klett-Fernald algorithm are quite good using both station data, showing differences below 12% at 355 and 532 nm for Cerro Poyos. In the case of the aerosol extinction profiles, good agreement was found between GRASP and the CAPS data (differences below 20%), while the comparison with PLASMA data shows larger differences. The SSA profiles show values typical of dust aerosols and the differences between retrievals using sun-photometer data measured at Granada

- and Cerro Poyos are below 10% at the lidar wavelengths. Other aerosol properties obtained with GRASP, like the color ratio and the backscatter Angström Exponent, also show similar values to those observed in the literature for dust aerosols. GRASP algorithm is quite robust as shows the agreement between the optical and microphysical properties retrieved by AERONET products and airborne measurements. Results obtained here show that the combination of lidar and sunphotometer data can provide improved and more complete column-integrated data compared to AERONET retrieval.

Reliable vertically-resolved properties such as the SSA, extinction or volume concentration are also provided, improving the capabilities of previous algorithms such as LIRIC.

Nonetheless, the retrieved scattering-Angström exponent profiles together with the better agreement found between Cerro Poyos retrievals and the aircraft compared to Granada retrievals indicate that GRASP vertical distribution of some of the aerosol properties is still affected by considerable uncertainties. This is somehow an expected result because of the use of the column-integrated sun-photometer data.

properties of aerosol layers with features really different than the atmospheric boundary layer aerosol. However, the

The analysis presented here is useful as a primary evaluation of the GRASP algorithm using sun-photometer and lidar signal to retrieve aerosol microphysical properties, both integrated along the vertical column and as vertical profiles. The use of a second sun-photometer located over the local atmospheric boundary layer can be very relevant for the study of the

20 presented analysis is representative of Saharan dust transport to south Europe and still it is necessary to use a more complete dataset that includes different aerosol loads and types. In future studies, we could try to use of the combination of one lidar with two sun-sky photometers at different height to try improve the retrievals in the cases with different aerosol layers. In addition, in order to validate the presented GRASP scheme, in the future it is planned to use global aerosol models (e.g. GEOS-5) following an approach similar to Whiteman et al., (2017).

## 25 Acknowledgements

This work was supported by the Andalusia Regional Government through project P12-RNM-2409, by the Spanish Ministry of Economy and Competitiveness through project CGL2013-45410-R, CGL2016-81092-R and through grant FPI (BES-2014-068893), also by "Juan de la Cierva-Formación" program (FJCI-2014-22052) and the Marie Skłodowska-Curie Individual Fellowships (IF) ACE\_GFAT (grant agreement No 659398) and by the University of Granada through "Plan

- Propio. Programa 9 Convocatoria 2013. The financial support for EARLINET in the ACTRIS Research Infrastructure Project by the European Union's Horizon 2020 research and innovation programme through project ACTRIS-2 (grant agreement No 654109). The authors thankfully acknowledge the FEDER program for the instrumentation used in this work and the Sierra Nevada National Park, for its support for the operation of Cerro Poyos station.
- This work is part of the ChArMEx project supported by ADEME, CEA, CNRS-INSU and Météo- France through the
   multidisciplinary programme MISTRALS (Mediterranean Integrated Studies aT Regional And Local Scales). We thank
   the instrument scientists, pilots and ground crew of SAFIRE for facilitating the instrument integration and conducting flight
   operations.

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

# Tables

| Parameter<br>measured | Instrument                                                                         | Abbreviation | Scientific objective            | Nominal size<br>range (µm) | Wavelength (nm) |
|-----------------------|------------------------------------------------------------------------------------|--------------|---------------------------------|----------------------------|-----------------|
|                       | Forward Scattering Spectrometer<br>Probe, Model 300, Particle<br>Measuring Systems | FSSP-300     | Coarse mode conc.               | 0.28 - 20                  | 632.8           |
| Size distribution     | Sky-Optical Particle Counter,<br>model 1.129, Grimm Technik                        | GRIMM        | Coarse mode conc.               | 0.25 - 32                  | 655             |
| Size distribution     | Ultra-High Sensitivity Aerosol<br>Spectrometer Droplet<br>Measurement Technologies | UHSAS        | Aiken + accumulation mode conc. | 0.04 - 1                   | 1054            |
|                       | Scanning mobility particle sizer,<br>custom-built                                  | SMPS         | Aiken + accumulation mode conc. | 0.03 - 0.4                 | n/a             |
|                       | 3λ Integrated Nephelometer,<br>Model 3563, TSI                                     | Nephelometer | Scattering coefficient          | n/a*                       | 450, 550, 700   |
| Optical properties    | Cavity Attenuated Phase Shift,<br>Aerodyne Research Inc.                           | CAPS         | Extinction coefficient          | n/a*                       | 530             |
|                       | Photomètre Léger Aéroporté pour<br>la Surveillance des Masses d'Air                | PLASMA       | Extinction coefficient,<br>AOD  | n/a*                       | 340-2250        |

 Table 1. Instruments on board the ATR-42 aircraft during F30 and F31 flights.

\*not applicable

## 5 Table 2. Input and output information used for LIRIC and GRASP retrievals.

|    | LIRIC                                  |                                         | GRASP                      |                        |  |
|----|----------------------------------------|-----------------------------------------|----------------------------|------------------------|--|
|    | <u>SUN-</u>                            | LIDAR                                   | SUN-PHOTOMETER             | LIDAR                  |  |
|    | PHOTOMETER*                            | Elastic backscattered                   | AOT or AOD                 | Elastic backscattered  |  |
| Ľ  | AOD                                    | signal:                                 | Total scattered            | signal:                |  |
| Π  | • VC                                   | • 355, 532 and 1064                     | radiances                  | • 355, 532 and         |  |
| A  | <ul> <li>RRI and IRI</li> </ul>        | nm                                      | At 440, 670, 870 and 1020  | 1064 nm                |  |
|    | % Sphericity                           | <ul> <li>532-cross polarized</li> </ul> | nm                         |                        |  |
|    |                                        | signal                                  |                            |                        |  |
|    | <ul> <li>VC profile for fit</li> </ul> | ne and coarse mode                      | Columnar (fine and coarse) | Vertical (fine and     |  |
|    |                                        |                                         | • SD                       | <u>coarse</u> )        |  |
| E  |                                        |                                         | RRI and IRI                | • VC                   |  |
| PC |                                        |                                         | • VC                       | • $\alpha$ and $\beta$ |  |
| IJ |                                        |                                         | • r <sub>eff</sub>         | • SSA                  |  |
| Ö  |                                        |                                         | • SSA                      |                        |  |
|    |                                        |                                         | • LR                       |                        |  |
|    |                                        |                                         | % Sphericity (total)       |                        |  |

\*AERONET product

**Table 3.** Columnar effective radius and particle volume concentration for coarse and fine particles modes retrieved by GRASP and AERONET algorithms.

|                                                                |         |        | 16 <sup>th</sup> June 2013 |             | 17 <sup>th</sup> June 2013 |             |
|----------------------------------------------------------------|---------|--------|----------------------------|-------------|----------------------------|-------------|
|                                                                |         |        | Granada                    | Cerro Poyos | Granada                    | Cerro Poyos |
| Effective<br>radius<br>(µm)                                    | GRASP   | fine   | 0.12                       | 0.13        | 0.10                       | 0.12        |
|                                                                |         | coarse | 2.2                        | 2.2         | 2.4                        | 2.2         |
|                                                                | AERONET | fine   | 0.12                       | 0.11        | 0.11                       | 0.12        |
|                                                                |         | coarse | 1.9                        | 1.9         | 2.1                        | 1.9         |
| Volume<br>concentration<br>(µm <sup>3</sup> /µm <sup>2</sup> ) | GRASP   | fine   | 0.018                      | 0.017       | 0.018                      | 0.011       |
|                                                                |         | coarse | 0.17                       | 0.15        | 0.14                       | 0.13        |
|                                                                | AERONET | fine   | 0.015                      | 0.015       | 0.016                      | 0.010       |
|                                                                |         | coarse | 0.14                       | 0.13        | 0.12                       | 0.11        |

**Table 4.** Comparison of fine, coarse and total mean volume concentration ( $\mu m^3/cm^3$ ) retrieved by GRASP, measured by airborne and retrieved by LIRIC for dust layers on 16<sup>th</sup> (up to 4.5 km a.s.l.) and 17<sup>th</sup> (from 2.6 to 5.0 km a.s.l.) June.

| Volume c              | Volume concentration |          | 16 <sup>th</sup> June 2013 17 <sup>th</sup> June 2013 |             | ine 2013      |
|-----------------------|----------------------|----------|-------------------------------------------------------|-------------|---------------|
| $(\mu m^3 / \mu m^2)$ |                      | Granada  | Cerro Poyos                                           | Granada     | Cerro Poyos   |
|                       | GRASP                | 4.7 ±0.6 | 5.5 ±0.3                                              | 5.5 ±1.3    | 3.5 ±1.0      |
| Fine                  | AIRBORNE             | 2.6 ±0.4 | 2.6 ±0.4                                              | 1.9 ±0.6    | 1.9 ±0.6      |
|                       | LIRIC                | 4.2 ±0.8 | 4.6 ±1.0                                              | 5.3 ±1.3    | $2.8 \pm 1.1$ |
| Coarse                | GRASP                | 28 ±4    | 32 ±4                                                 | 35 ±7       | 36 ±5         |
|                       | AIRBORNE             | 31 ±8    | 27 ±5                                                 | 41 ±11      | 41 ±11        |
|                       | LIRIC                | 37 ±4    | 46 ±4                                                 | 35 ±5       | 38 ±6         |
| Total                 | GRASP                | 33 ±4    | 38 ±4                                                 | 40 ±8       | 39 ±6         |
|                       | AIRBORNE             | 33 ±8    | 28 ±5                                                 | $42 \pm 11$ | 42 ±11        |
|                       | LIRIC                | 41 ±5    | 50 ±4                                                 | 40 ±6       | 41 ±6         |

5 **Table 5.** Mean value of backscatter - Angström exponent (β-AE) and Color Ratio (CR) between 532 and 1064 nm, retrieved by GRASP for dust layers on 16<sup>th</sup> and 17<sup>th</sup> June 2013.

|      | 16 <sup>th</sup> June 2013 |                 | 17 <sup>th</sup> June 2013 |            |
|------|----------------------------|-----------------|----------------------------|------------|
|      | Granada                    | Granada         | Granada                    | Granada    |
| β-ΑΕ | 0.65 ±0.07                 | $0.46 \pm 0.05$ | 0.63 ±0.12                 | 0.40 ±0.10 |
| CR   | 1.15 ±0.05                 | 1.13 ±0.05      | $1.09 \pm 0.09$            | 1.08 ±0.04 |

## Figures