# Peer review of "Comparative assessment of GRASP algorithm for a dust event over Granada (Spain) during ChArMEx-ADRIMED 2013 campaign."

_Atmospheric Measurement Techniques, 2017_

## Referee Comment (RC1) · Anonymous Referee #1 · 21 Jul 2017

Review for manuscript Comparative assessment of GRASP algorithm for a dust event over Granada (Spain) during ChArMEx-ADRIMED 2013 campaign. Authors provide comparison of inversions of lidar and sun photometers observations using three different algorithms: LIRIC, GARLIC, AERONET operational algorithm, and demonstrate that results are similar. Such comparison is useful, showing that approaches are consistent. On another hand, similarity in results is hardly surprising, because all three algorithms are based on the same principles. Possibility to use two sun photometers at different heights is interesting, because it helps to analyze possible biases due to geometrical overlap effects. I think manuscript can be published after some revision.

[Figure]

General comments The main question is what we can conclude from this comparison? Authors write: "Results obtained here show that the combination of lidar and sun photometer data can provide improved and more complete column-integrated data compared to AERONET retrieval." I think this statement is unsupported. The difference between methods is inside the inversion uncertainty. This is just comparison and can not be considered as validation. In conclusion they write: "As a future outlook, it will be of great interest to expand the present analysis covering different scenarios including a major variety of aerosol types and loads during campaigns with airborne measurements in order to validate the new improvements". Yes, it is always useful to consider more situations; still it is not validation. Expected advantage of combining lidar with sun photometers is ability to profile intensive particle properties, such as effective radius, refractive index, Angstrom exponent. Authors provide profiles of volume and backscattering, so it is difficult to conclude if they observe height dependence of intensive parameters. Authors write "For 17th June, vertical profiles of SSA are sensitive to the different aerosol layers with different aerosol types illustrating the capabilities of GRASP for detecting different aerosol layers with different composition." But from fig.7, 8 I can conclude in the height range ∼1.8 – 2.7 km backscattering is very low, so variation of SSA in this range is probably just artifact. The same is true for fig.9, variations of AE in this range are probably not real. Do authors have depolarization measurements? Height variation of particle depolarization ratio could provide some information. Specific comments 1. Fig.3. In Granada imaginary part has spectral dependence typical for dust, while In Cerro Poyos no. Why? PSD look similar. Is it possible to provide vertical profile of imaginary part? 2. Information about airplane measurements would be useful. Did it ascend by spiral? How much time did it take for one vertical profile? 3. Fig.6, 17 June, Granada, 355 nm. Why Klett is not given for ∼1.8 - 2.5 km? If it is 0, it still should be shown. Why Klett at 355 is not shown below 1.6 km while Grasp retrievals are given? The same questions are for Cerro Poyos. 4. References take about 50% of the text volume. Probably too much.

---

## Referee Comment (RC2) · Anonymous Referee #2 · 14 Aug 2017

General comments: This paper presents the comparative assessment of the GRASP (Generalized Retrieval of Atmosphere and Surface Properties) algorithm aerosol optical properties using combined sunphotometer and lidar measurements with other ground-based lidar products and airborne-based in-situ measurement during ChArMEx-ADRIMED 2013 campaign. The second section of the paper is the explanation of Granada site and instrumentation of ground-based remote sensing (sunphotometer and lidar) and airborne in-situ measurements during the campaign. The list of equipment, retrieved/measured optical properties with algorithm characteristics, and its uncertainties are presented with references. The third section is the explanation of GRASP and LIRIC inversion algorithms. Although both algorithms use the information

of combined lidar and sun-sky photometer measurements, the detailed processes are different and described in this section. The main advantage of GRASP is the simultaneous inversion using 180° backscattering information of lidar and direct-sun and almucantar measurement of sky radiance by sun-sky radiometer. The fourth section contains the comparison results of GRASP and other measurements according to different optical properties: column-integrated properties such as size distribution, effective radii of course and fine modes, volume concentration, refractive indices, and single scattering albedo, and vertically-resolved properties such as volume concentration, extinction and backscatter coefficient profiles, single scattering albedo, and scattering Ångström Exponent. Most aerosol optical properties of GRASP show similar results with other measurements, but also provide different information such as coarse mode shift to higher radii compared to AERONET-only or more information such as profiles of SSA and scattering Ångström Exponent compared to LIRIC.

The paper presents an abundant comparison results of GRASP with other measurement during the campaign, and the results are clear. The scope is well-addressed also, thus I recommend it for publication after the responses for some points listed hereafter.

Specific comments/questions:

1) In section 2.1, the distribution of observation sites or geological map could help to understand geographical conditions although the located information is described in manuscript because the authors explain apparent errors as the different location of measurement sites at some compared aerosol optical properties.

2) In section 2.2 (page 4 line 16), recent AERONET data version is changed from 2 to 3. Although the version of inversion data is still version 2, please notate the data version (i.e. version 2).

3) In section 3, please describe the input and output information of lidar and sun-photometer for LIRIC and GRASP specifically (i.e. which wavelength of sky-radiance of sunphotometer, lidar measurement as input, and which column-integrated/vertical

aerosol optical properties as output).

4) In section 3 (page 5 line 31), Which information of the photometer is provided for lidar retrievals? Please describe "otherwise" (the assumption in lidar measurements) more specifically.

5) In section 4 (figure 1), what is the definition of the lidar range corrected signal? Is it calculated as P*r2, where P is the lidar data (received power) and r is the range?

6) In section 4.1 (page 7 line 11 and 27): the authors mentioned that the wavelength of RRI and IRI from airborne measurement is 500 nm. However, it is not on the 500 nm in the Figure 3. Please clarify it.

7) In section 4.1, which height is represented from the airborne measurement? Would it be possible reason of difference because lidar profile shows different concentration in 17 June?

8) In section 4.1 (page 7 line 32) which wavelength for SSA of "0.80-0.90"?

9) In section 4.2 (page 8 line 30), please briefly describe difference in the method of "Klett" and "GRASP" in terms of lidar ratio (LR).

10) In section 4.2, (page 8 line 39), could you explain why the discrepancies b/w two B-coeffi products are getting larger in longer wavelength although the error range of GRASP B-coeffi profile is smaller in longer wavelength in Granada on 16th June and Cerro Poyos on 17th June?

11) In section 4.3 (page 9 line 12), could you show the error range of SSA profiles from GRASP? It could be useful information to understand this SSA profiles because the AODs of all cases are less than 0.3 and SSA error could be large.

12) In section 5 (page 10 line 20), I agree for the combination of lidar and sunphotometer data in GRASP algorithm can provide improved and more complete data compared to AERONET retrieval. The refractive indices of GRASP show better agreement

with in-situ measurement compared to those of AERONET. However, the column-integrated size distribution and SSA doesn't show any in-situ measurement results together. Could you quantify the improvement in GRASP compared to AERONET measurement?

13) In Section 5 (page 10 line 29-30), please explain which the improvements of GRASP are new and the method of a second sun-sky photometer in your plan. Is the three instruments combination as one lidar and two sun-photometers in GRASP?

Technical correction:

Please check whether the typos I found are correct.

1) In section 2.1 (page 3 line 31) it local sources → its local sources

2) In section 2.2 (page 4 line 20), removing ")".

3) In section 3 (page 5 line 29): lidar "an" sun-sky photometer measurements → lidar "and" sun-sky ..

4) In section 3 (page 5 line 35), please write full name of "CRI" (is it "components of refractive indices"?)

5) In section 5 (page 10 line 29), vaity → variety
* * *

---

## Editor Comment (EC1) · F. Dulac (Editor) · 30 Aug 2017

I encourage a revision that carefully addresses reviewers comments, especially the strong criticisms raised by referee #1.

Technical corrections: in the conclusion, specifiy "complete lidar overlap range" in the begining of line 4 ; check uodated reference for Torres et al. paper submitted to ACP, which is now in press; please use larger characters and bold style in order to improve the readability of axes legends and box legends in figures 2-4.
* * *

---

## Author Comment (AC1) · 18 Sep 2017

**Response to comments #1**

We would like to thank the referee #1 for their constructive and useful comments. This document contains the authors' responses to comments from reviewer #1. Each comment is discussed separately with the following typesetting:

> **\*Reviewer's comment**
>
> **Author's response**
>
> Changes in the manuscript.

*Review for manuscript Comparative assessment of GRASP algorithm for a dust event over Granada (Spain) during ChArMEx-ADRIMED 2013 campaign. Authors provide comparison of inversions of lidar and sun photometers observations using three different algorithms: LIRIC, GARLIC, AERONET operational algorithm, and demonstrate that results are similar. Such comparison is useful, showing that approaches are consistent. On another hand, similarity in results is hardly surprising, because all three algorithms are based on the same principles. Possibility to use two sun photometers at different heights is interesting, because it helps to analyze possible biases due to geometrical overlap effects. I think manuscript can be published after some revision.

**General comments:**

*The main question is what we can conclude from this comparison? Authors write: "Results obtained here show that the combination of lidar and sun photometer data can provide improved and more complete column-integrated data compared to AERONET retrieval." I think this statement is unsupported. The difference between methods is inside the inversion uncertainty. This is just comparison and can not be considered as validation.

**The comparison presented here between GRASP versus other algorithms (i.e. AERONET and LIRIC) and with airborne in-situ measurements shows the potential to retrieve aerosol microphysical and optical profiles, and also to obtain fine and coarse mode aerosol refractive index and single scattering albedo which is not possible with the current AERONET inversion. Also, the advance versus LIRIC is that GRASP does not assumes as starting point the results of AERONET inversion.**

*In conclusion they write: "As a future outlook, it will be of great interest to expand the present analysis covering different scenarios including a major variety of aerosol types and loads during campaigns with airborne measurements in order to validate the new improvements". Yes, it is always useful to consider more situations; still it is not validation.

**We agree with the referee that the results presents are just an evaluation for two study cases. To perform evaluations of the configuration sun-photometer + lidar signals, we plan to use a synthetic database provided by global models. Initially we will work with GEOS-5 aerosol fields and computing sky radiances using VLDORT radiative transfer code. A complete evaluation using this scheme is in course, but outside the scope of this manuscript.**

We add (section 5, page 11, line 19): The analysis presented here is useful as a primary evaluation of the GRASP algorithm using sun-photometer and lidar signal to retrieve aerosol microphysical properties, both integrated along the vertical column and as vertical profiles. The use of a second sun-photometer located over the local atmospheric boundary layer can be very relevant for the study of the properties of aerosol layers with features really different than the atmospheric boundary layer aerosol. However, the presented analysis is representative of Saharan dust transport to south Europe and still it is necessary to use a more complete dataset that includes different aerosol loads and types. In future studies, we could try to use of the combination of one lidar with two sun-sky photometers at different height to try improve the retrievals in the cases with different aerosol layers. In addition, in order to validate the presented GRASP scheme, in the future it is planned to use global aerosol models (e.g. GEOS-5) following an approach similar to Whiteman et al., (2017).

*Expected advantage of combining lidar with sun photometers is ability to profile intensive particle properties, such as effective radius, refractive index, Angstrom exponent. Authors provide profiles of volume and backscattering, so it is difficult to conclude if they observe height dependence of intensive parameters.

**The focus of this work was to evaluate GRASP retrieval algorithm against the well-established AERONET inversion retrievals and independent in-situ airborne measurements during Charmex campaign. GRASP is in continuous development and there are intensive parameters profiles that the referee mentions that are not still available and therefore their evaluation is not possible. However, we focused on SSA and Scattering – Angström Exponent, and also other intensive properties such as mean value of backscatter - Angström exponent (β-AE) and Color Ratio (CR).**

*Authors write "For 17th June, vertical profiles of SSA are sensitive to the different aerosol layers with different aerosol types illustrating the capabilities of GRASP for detecting different aerosol layers with different composition." But from fig.7, 8 I can conclude in the height range ~1.8 – 2.7 km backscattering is very low, so variation of SSA in this range is probably just artifact. The same is true for fig.9, variations of AE in this range are probably not real. Do authors have depolarization measurements? Height variation of particle depolarization ratio could provide some information.

**We agree with the reviewer that in the mentioned layer the aerosol load is low (~5 $\mu m^3/cm^3$ in the range ~1.8 – 2.7 km a.s.l.) and, hence, SSA and AE values could be affected by a large uncertainty. However, the layer below 1.8 km a.s.l.**

showed a moderate concentration (~17 $\mu m^3/cm^3$) and the SSA and AE profiles still reveal a different composition to that of the layer above 2.7 km a.s.l..

We add (section 4.2, page 10, line 21): On 17th June, in the range ~1.8 – 2.7 km a.s.l. the aerosol load was low (~5 µm³/cm³) and, hence, SSA and AE values could be affected by larger uncertainties. However, the layer up to 1.8 km a.s.l. showed a moderate concentration (~17 µm³/cm³) and the SSA and AE profiles still reveal a different composition with a different composition to that of the layer above 2.7 km a.s.l.

We added the depolarization ratio ($\delta$) quicklook to Figure 2. These figures point out that on 16th June there was a unique layer while on 17th June, there were two main layers: an aerosol layer close to the surface and a decoupled one between 2.7 and 5.5 km a.s.l.. The depolarization ratio evidences that there was an aerosol type below 2.7 km a.s.l. and a different one above this altitude up to 5.5 km a.s.l..

We add (section 4, page 6, line 13): "measurements of $\delta$ evidence that there was an aerosol layer below 2.7 km a.s.l. and another aerosol layer above this altitude up to 5.5 km a.s.l.."

[Figure]

**Figure 1.** Temporal evolution of the lidar range corrected signal (top) and the depolarization ratio (bottom) at 532nm on 16th (left) and 17th (right) June, 2013. The two purple lines indicate the lidar analyzed interval. The black dashed line indicates the sun-photometer measurements.

**Specific comments:**

*Fig.3. In Granada imaginary part has spectral dependence typical for dust, while In Cerro Poyos no. Why? PSD look similar. Is it possible to provide vertical profile of imaginary part?

**We thank the referee for this comment as we did not notice before. We added the following comment in the manuscript. For the moment, GRASP do not provide vertical profiles of refractive indices.**

We add (section 4.1, page 7, line 28): At Cerro Poyos we did not find the spectral dependence of the IRI typically associated to mineral dust. The AOD at 440 nm were around 0.18 - 0.27 and we used AERONET level 1.5 products, therefore, these values have large uncertainties (> 50%; Dubovik et al., 2000). The lack of spectral dependence can be just an artifact of the inversion. However, it is worthy to note that at Cerro Poyos the PSD shows a mode in the coarse mode size range around 1 μm. As there is still discussion in the scientific community about dust RI and about the differences in dust particles between different sources (e.g. Colarco et al., 2014), results can suggest possible differences in dust RI between long range transported and mixture with local dust injections (the area is very dry in summer, thus favoring local mineral dust resuspension) and local pollution.

*Information about airplane measurements would be useful. Did it ascend by spiral? How much time did it take for one vertical profile?

**During the CHARMEX campaign the flights ascended or descended performing a spiral trajectory during 30 min. We have added in Figure 2 the flight time for both days.**

We add (section 2.3, page 5, line 1): These flights ascended or descended performing a spiral trajectory during 30 min.

*Fig.6, 17 June, Granada, 355 nm. Why Klett is not given for ~1.8 - 2.5 km? If it is 0, it still should be shown. Why Klett at 355 is not shown below 1.6 km while Grasp retrievals are given? The same questions are for Cerro Poyos.

**On 17[th] June for this range at 355nm the values were 0. We changed the figures and now we show these values.**

We add (section 4.2, page 9, line 17): Below 1.6 km, the Klett retrieval at 355 showed unrealistic values probably associated with instrumental problems. However, for GRASP this problem does not appear and seems to be canceled due to the use of the combined data of lidar and sun-photometer.

*References take about 50% of the text volume. Probably too much.

**In the revised manuscript, we have reduced the number of references.**

---

## Author Comment (AC2) · 18 Sep 2017

**Response to comments #2**

We would like to thank the referee #2 for their constructive and useful comments. This document contains the authors' responses to comments from reviewer #2. Each comment is discussed separately with the following typesetting:

*Reviewer's comment

**Author's response**

Changes in the manuscript.

*General comments: This paper presents the comparative assessment of the GRASP (Generalized Retrieval of Atmosphere and Surface Properties) algorithm aerosol optical properties using combined sunphotometer and lidar measurements with other ground-based lidar products and airborne-based in-situ measurement during ChArMEx-ADRIMED 2013 campaign. The second section of the paper is the explanation of Granada site and instrumentation of ground-based remote sensing (sunphotometer and lidar) and airborne in-situ measurements during the campaign. The list of equipment, retrieved/measured optical properties with algorithm characteristics, and its uncertainties are presented with references. The third section is the explanation of GRASP and LIRIC inversion algorithms. Although both algorithms use the information of combined lidar and sun-sky photometer measurements, the detailed processes are different and described in this section. The main advantage of GRASP is the simultaneous inversion using 180◦ backscattering information of lidar and direct-sun and almucantar measurement of sky radiance by sun-sky radiometer. The fourth section contains the comparison results of GRASP and other measurements according to different optical properties: column-integrated properties such as size distribution, effective radii of course and fine modes, volume concentration, refractive indices, and single scattering albedo, and vertically-resolved properties such as volume concentration, extinction and backscatter coefficient profiles, single scattering albedo, and scattering Ångström Exponent. Most aerosol optical properties of GRASP show similar results with other measurements, but also provide different information such as coarse mode shift to higher radii compared to AERONET-only or more information such as profiles of SSA and scattering Ångström Exponent compared to LIRIC.

The paper presents an abundant comparison results of GRASP with other measurement during the campaign, and the results are clear. The scope is well-addressed also, thus I recommend it for publication after the responses for some points listed hereafter.

**Specific comments/questions:**

*In section 2.1, the distribution of observation sites or geological map could help to understand geographical conditions although the located information is described in manuscript because the authors explain apparent errors as the different location of measurement sites at some compared aerosol optical properties.

**We agree with the referee and we have added a map illustrating the Granada and Cerro Poyos stations.**

We add in the manuscript (section 2.1, page 3, line 21): Figure 1 shows a map illustrating the distance between Granada and Cerro Poyos stations.

We add in the manuscript (section 2.3, page 5, line 2): Figure 1 shows the spiral trajectory of F31 flight that is similar to that of F30, covering in both cases the same atmospheric column.

[Figure]

**Figure 1.** Map illustrating the Granada and Cerro Poyos stations. The red line indicates the trajectory and the black points the altitude of the aircraft on 17[th] June

*In section 2.2 (page 4 line 16), recent AERONET data version is changed from 2 to 3. Although the version of inversion data is still version 2, please notate the data version (i.e. version 2).

**We added the AERONET data version used in this paper (version 2).**

We changed (section 2.2, page 4, line 17): "In this work, the AERONET Version 2 Level 2.0 data obtained at Granada and Cerro Poyos during ChArMEx-ADRIMED 2013 are used."

* In section 3, please describe the input and output information of lidar and sunphotometer for LIRIC and GRASP specifically (i.e. which wavelength of sky-radiance of sunphotometer, lidar measurement as input, and which column-integrated/vertical aerosol optical properties as output).

**We add a new table with input/output information used/retrieved by GRASP and LIRIC.**

We add the following sentence (section 3, page 5, line 21): "The input information needed by GRASP and LIRIC algorithms and the aerosol properties retrieved and used in this work are shown in Table 2".

**Table 2.** Input and output information used for LIRIC and GRASP retrievals.

| | LIRIC | | GRASP | |
|---|---|---|---|---|
| | SUN-PHOTOMETER* | LIDAR | SUN-PHOTOMETER | LIDAR |
| **INPUT** | • AOD
• VC
• RRI and IRI
• % Sphericity | Elastic backscattered signal:
• 355, 532 and 1064 nm
• 532-cross polarized signal | • AOT or AOD
• Total scattered radiances
At 440, 670, 870 and 1020 nm | Elastic backscattered signal:
• 355, 532 and 1064 nm |
| **OUTPUT** | • VC profile for fine and coarse mode | | Columnar (fine and coarse)
• SD
• RRI and IRI
• VC
• $r_{eff}$
• SSA
• LR
• % Sphericity (total) | Vertical (fine and coarse)
• VC
• $\alpha$ and $\beta$
• SSA |

*AERONET product

*In section 3 (page 5 line 31), Which information of the photometer is provided for lidar retrievals? Please describe "otherwise" (the assumption in lidar measurements) more specifically.

**The sun-photometer provides the aerosol optical depth and the combination of direct sun and sky radiances provides column-integrated aerosol microphysical properties. Sun-photometer measurements, collocated with backscatter lidar, are used to estimate extinction-to-backscatter ratio, typically known as lidar ratio (LR). If there are no collocated sun photometer, the LR assumption are based on climatological values.**

We changed (section 3, page 5, line 35): "… and the photometer data provides the information (e.g. amount and type) required for lidar retrievals that otherwise would be assumed from climatological data (Bovchaliuk et al., 2016)"

*In section 4 (figure 1), what is the definition of the lidar range corrected signal? Is it calculated as P*r2, where P is the lidar data (received power) and r is the range?

**Yes, the referee is right.**

We add the following sentence (section 4, page 6, line 9): The RCS is calculated as P*r2, where P is the lidar signal (corrected from background and dark current) and r is the altitude.

*In section 4.1 (page 7 line 11 and 27): the authors mentioned that the wavelength of RRI and IRI from airborne measurement is 500 nm. However, it is not on the 500 nm in the Figure 3. Please clarify it.

**We made a mistake in the text but not in figure 3. The wavelength of RRI and IRI from airborne measurements is 530 nm and not 500 nm. We have changed it in section 4.1.**

*In section 4.1, which height is represented from the airborne measurement? Would it be possible reason of difference because lidar profile shows different concentration in 17 June?

**We think that the referee refers to section 4.2 indeed, where we use airborne data. Height is above sea level (a.s.l.) which is the same than for lidar data.**

**We believe that the differences in particle volume concentrations between GRASP and airborne data below 2km a.s.l on 17th June are explained because the flight was not exactly over Granada city as shown in Fig. 1 and in the firsts kilometers of the atmosphere could be more differences in the horizontal.**

We add (section 4.2, page 9, line 4): On 17th June for Granada retrieval, the differences between both algorithms and airborne data below 2 km a.s.l. could be explained because the flight was not exactly over Granada city as shown in Fig. 1 and in the first two kilometers of the atmosphere differences are expected because of the influence of the city.

*In section 4.1 (page 7 line 32) which wavelength for SSA of "0.80-0.90"?

**Thanks to the referee comments we notice that there was a misspelling in the current version of the manuscript.**

We changed this sentence in the text (section 4.1, page 8, line 14): "The retrieved SSA values are in the range between 0.85-0.98 (355-1064 nm wavelength range) and are in the ranges…"

*In section 4.2 (page 8 line 30), please briefly describe difference in the method of "Klett" and "GRASP" in terms of lidar ratio (LR).

**The Klett uses the assumption of constant LR for the entire column. LR ratio were computed fitting the integral of the extinction to the measured aerosol optical depth. However, GRASP retrieves the LR (fine and coarse) directly from the sunphotometer and lidar measurements. The text has been modified in the methodology section to clarify these points.**

We add (section 4.2, page 9, line 12): The LR used in Klett method is assumed constant for the entire profile and was computed by fitting the integral of the different extinction profiles to the measured aerosol optical depth. However, GRASP uses both sun/sky radiances and the backscatter lidar data to provide LR values, both in column-integrated and vertical profiles.

*In section 4.2, (page 8 line 39), could you explain why the discrepancies b/w two B-coeffi products are getting larger in longer wavelength although the error range of GRASP B-coeffi profile is smaller in longer wavelength in Granada on 16th June and Cerro Poyos on 17th June?

**Only two study cases are presented here, so we cannot assume that discrepancies are larger for longer wavelengths or that the error range is smaller for a certain wavelength. An additional study, with a synthetic database provided by global models and an analysis with more cases, would be needed to get to significant conclusions in this respect. In the cases presented here, the discrepancies between the backscatter coefficient retrieved by Klett and GRASP were within the uncertainties for our system (~30%).**

*In section 4.3 (page 9 line 12), could you show the error range of SSA profiles from GRASP? It could be useful information to understand this SSA profiles because the AODs of all cases are less than 0.3 and SSA error could be large.

**GRASP developers are currently doing in depth studies of errors in SSA, and therefore, we are not able to provide them. That was commented in section 3.**

*In section 5 (page 10 line 20), I agree for the combination of lidar and sunphotometer data in GRASP algorithm can provide improved and more complete data compared to AERONET retrieval. The refractive indices of GRASP show better agreement with in-situ measurement compared to those of AERONET. However, the column integrated size distribution and SSA doesn't show any insitu measurement results together. Could you quantify the improvement in GRASP compared to AERONET measurement?

**We cannot compare the column-integrated size distribution between GRASP, AERONET and in-situ measurements because the latter are not available. For to quantify the improvement of GRASP compared to AERONET would be**

**necessary study more cases with more in-situ measurements at different wavelengths. However, we add the column-integrated SSA value measured in-situ in Figure 5.**

We modified the text (section 4.1, page 8, line 8): Moreover, the SSA value at 530 nm calculated by Denjean et al. (2016) for dust layer using airborne measurements during the campaign was 0.95 ±0.04. SSA retrieved by GRASP at 532 nm are close to the airborne value. Better agreement with this value is found for the retrievals from Granada on 16[th] June and at Cerro Poyos on 17[th] June. The differences from Granada on 17[th] June could be due the in-situ value was calculated for the dust layer whereas that GRASP and AERONET use sun-photometer data, which measures the total atmospheric column. Furthermore, in the case of Granada station, these measures could be influenced by injections of local pollution.

[Figure]

**Figure 5.** Single-scattering albedo retrieved by GRASP (blue) with its uncertainty (shaded area), AERONET (green) and airborne measurement for dust layer (black) on 16[th] (top) and 17[th] (bottom) June 2013 at Granada (left) and Cerro Poyos (right).

\* In Section 5 (page 10 line 29-30), please explain which the improvements of GRASP are new and the method of a second sun-sky photometer in your plan. Is the three instruments combination as one lidar and two sun-photometers in GRASP?

**GRASP exploits the synergy among lidar and sun-photometer data just providing enhanced column integrated retrievals over the standard AERONET approach and at the same time allows a profiling of some aerosol microphysical**

**properties. The use of a second sun-photometer located over the local atmospheric boundary layer can be very relevant for the study of the properties of aerosol layers with features really different than the atmospheric boundary layer aerosol. Thus, the use of the second photometer can be crucial for monitoring microphysics of long range transported particles in the free troposphere. In response to referee question: Yes, in the future, we could try to use of the combination of one lidar with two sun-sky photometers at different height.**

**We decided delete this sentence and we add the following conclusion:**

We add (section 5, page 11, line 19): The analysis presented here is useful as a primary evaluation of the GRASP algorithm using sun-photometer and lidar signal to retrieve aerosol microphysical properties, both integrated along the vertical column and as vertical profiles. The use of a second sun-photometer located over the local atmospheric boundary layer can be very relevant for the study of the properties of aerosol layers with features really different than the atmospheric boundary layer aerosol. However, the presented analysis is representative of Saharan dust transport to south Europe and still it is necessary to use a more complete dataset that includes different aerosol loads and types. In future studies, we could try to use of the combination of one lidar with two sun-sky photometers at different height to try improve the retrievals in the cases with different aerosol layers. In addition, in order to validate the presented GRASP scheme, in the future it is planned to use global aerosol models (e.g. GEOS-5) following an approach similar to Whiteman et al., (2017).

**Technical correction:**

Please check whether the typos I found are correct.

*In section 2.1 (page 3 line 31) it local sources → its local sources

We have changed "it local sources" by "its local sources" as suggested.

*In section 2.2 (page 4 line 20), removing ")".

We have removed ")" as suggested.

*In section 3 (page 5 line 29): lidar "an" sun-sky photometer measurements → lidar "and" sun-sky.

We have changed "lidar 'an' sun-sky photometer measurements" by "lidar 'and' sun-sky" as suggested.

*In section 3 (page 5 line 35), please write full name of "CRI" (is it "components of refractive indices"?)

CRI is Complex Refractive Indices. Due to that CRI is used only one time in the paper we decided changed it by "RRI, IRI" (real and imaginary refractive indices).

*In section 5 (page 10 line 29), vaity → variety

This word does not appear now.

---

## Author Comment (AC3) · 18 Sep 2017

**Response to comments #3**

We would like to thank to the editor Dr. Dulac for their constructive and useful comments. This document contains the authors' responses to comments from the editor. Each comment is discussed separately with the following typesetting:

**\*Reviewer's comment**

**Author's response**

Changes in the manuscript.

\*I encourage a revision that carefully addresses reviewers comments, especially the strong criticisms raised by referee #1.

**Technical corrections:**

\*In the conclusion, specifiy "complete lidar overlap range" in the begining of line 4

We add (section 5, page 10, line 31): Data from a second photometer at 1.2 km above the lidar system are also used, in this way avoid problems due to incomplete overlap.

\*Check uodated reference for Torres et al. paper submitted to ACP, which is now in press;

**The reference for Torres et. al. (2016) submitted to AMT is what we have in the manuscript. We changed this reference by the following:**

Torres, B., Dubovik, O., Fuertes, D., Schuster, G., Cachorro, V. E., Lapionak, T., Goloub, P., Blarel, L., Barreto, A., Mallet, M., Toledano, C., and Tanré, D.: Advanced characterization of aerosol properties from measurements of spectral optical depth using the GRASP algorithm, Atmos. Meas. Tech., Accepted. 2017.

\*Please use larger characters and bold style in order to improve the readability of axes legends and box legends in figures 2-4.

**We improved the readability of the figures and modified them in the revised manuscript.**

---

## Author Comment (AC4) · 18 Sep 2017

The referee can find attached the revised manuscript.

Please also note the supplement to this comment:
https://www.atmos-meas-tech-discuss.net/amt-2017-200/amt-2017-200-AC4-supplement.pdf
* * *